# Distil-E2D: Distilling Image-to-Depth Priors for Event-Based Monocular Depth Estimation

**Jie Long Lee**     **Gim Hee Lee**

Department of Computer Science, National University of Singapore

jielong.lee@u.nus.edu, gimhee.lee@nus.edu.sg

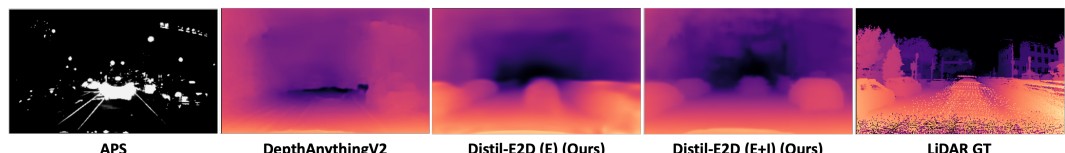

Figure 1: Monocular depth estimation under low-light conditions. An image-based monocular depth estimator (DepthAnythingV2) fails to recover meaningful structure from a nighttime APS frame due to poor lighting. In contrast, our event-based Distil-E2D produces coherent depth predictions, highlighting the advantages of event sensing and our proposed depth prior transfer framework.

## Abstract

Event cameras are neuromorphic vision sensors that asynchronously capture pixel-level intensity changes with high temporal resolution and dynamic range. These make them well suited for monocular depth estimation under challenging lighting conditions. However, progress in event-based monocular depth estimation remains constrained by the *quality* of supervision: LiDAR-based depth labels are inherently sparse, spatially incomplete, and prone to artifacts. Consequently, these signals are suboptimal for learning dense depth from sparse events. To address this problem, we propose Distil-E2D, a framework that distills depth priors from the image domain into the event domain by generating dense synthetic pseudolabels from co-recorded APS or RGB frames using foundational depth models. These pseudolabels complement sparse LiDAR depths with dense semantically rich supervision informed by large-scale image-depth datasets. To reconcile discrepancies between synthetic and real depths, we introduce a Confidence-Guided Calibrated Depth Loss that learns nonlinear depth alignment and adaptively weights supervision by alignment confidence. Additionally, our architecture integrates past predictions via a Context Transformer and employs a Dual-Decoder Training scheme that enhances encoder representations by jointly learning metric and relative depth abstractions. Experiments on benchmark datasets show that Distil-E2D achieves state-of-the-art performance in event-based monocular depth estimation across both event-only and event+APS settings. Code available at the project website[1].

## 1   Introduction

Event cameras are neuromorphic vision sensors that asynchronously detect per-pixel brightness changes with microsecond precision and high dynamic range. Unlike conventional cameras, they are

---

[1]https://github.com/leejielong/Distil-E2D

39th Conference on Neural Information Processing Systems (NeurIPS 2025).

more resilient to motion blur and perform well in extreme lighting, enabling low-latency, high-speed vision. These properties of event cameras make them ideal for 3D perception tasks such as monocular depth estimation (MDE). Event-based monocular depth estimation (EMDE) provides an efficient and resilient approach to perceiving scene geometry under conditions involving high-speed motion, low illumination, and extreme dynamic range. EMDE applications span diverse domains, including autonomous driving [1–4], aerial robotics [5], robotic manipulation [6], and low-latency spatial reasoning in AR/VR systems [7]. These applications highlight the versatility and growing relevance of EMDE in real-time, real-world perception.

EMDE aims to recover dense depth from single-view event streams—an inherently ill-posed problem due to the absence of multi-view geometric constraints. Event data lack standard image-based depth cues such as absolute intensity, texture, shading, and color gradients. Additionally, the sparsity of events driven by scene and camera motion leads to uneven spatial coverage especially in static or low-motion regions. These factors cause EMDE to be significantly more challenging than its image-based counterpart. Recent learning-based approaches [8–12] have shown promising results in predicting depth directly from event streams. However, the quality of supervision remains a fundamental bottleneck. Most event-based depth datasets [3, 2] rely on LiDAR scans as ground truth, which poses several intrinsic challenges due to mismatched sensing modalities. LiDAR acquires depth via discrete angular laser scans to produce point clouds that give spatially sparse supervision. Furthermore, LiDAR utilizes mechanical scanning that integrates depth over a short but non-negligible temporal window. This temporal ambiguity introduces a mismatch with the microsecond precision of event data that complicates accurate event-depth association. Additionally, in dynamic scenes, the sequential scanning process introduces scanline artifacts—parallel streaks of occlusive depth measurements. Collectively, spatial sparsity, temporal misalignment, and scanline artifacts of LiDAR limit its effectiveness for training dense and temporally precise EMDE models.

Efforts to overcome supervision limitations through simulated data generation [8, 9] offer an alternative by simulating both event streams and dense depth from virtual environments. These simulated datasets provide a scalable means to generate diverse scenes with pixel-perfect labels, which circumvent the hardware constraints and annotation challenges of real-world data collection. However, they come with significant drawbacks. The simulated scenes often lack the visual complexity, noise characteristics, and motion dynamics of real-world environments. Moreover, simplified event generation models are used in simulation, resulting in event distributions that deviate from real-world event statistics. This introduces domain gaps in both appearance and signal distribution, which substantially degrades the generalization of models trained on simulated data when deployed in real-world settings.

On the other hand, image-based MDE has progressed rapidly in recent years, giving rise to foundational depth models [13–19] with strong zero-shot generalization to diverse scenes. These foundational models are propelled by access to large-scale RGB-depth datasets [20–23] and robust self-supervised learning frameworks. These models learn strong geometric and semantic priors, enabling dense depth prediction with high spatial consistency and generalization across a wide range of environments. Despite their success, these rich priors have yet to be exploited in event-based settings to overcome supervision limitations.

In this work, we address the supervision gap in event-based monocular depth estimation by proposing Distil-E2D, a novel distillation framework that transfers dense depth priors from foundational models into the event domain. Our key idea is to leverage co-recorded intensity images to generate high-quality synthetic depth pseudolabels using image-based foundational depth models. These pseudolabels provide dense and semantically rich supervision that complements sparse and noisy LiDAR ground truth for more effective learning of spatial structure from events. However, due to modality differences, synthetic depths exhibit scale and alignment discrepancies with LiDAR. To mitigate this, we introduce the Confidence-guided Calibrated Depth Loss (CCDL), comprising: 1) a Nonlinear Depth Calibration (NDC) module that learns a mapping to align synthetic and LiDAR depths, and 2) an Alignment-aware Confidence Estimator (ACE) that generates a pixel-wise confidence map by assessing local consistency to emphasize reliable supervision. Additionally, we propose a novel architecture featuring a Context Transformer (CT) to integrate past prediction context via cross-attention, and a Dual-Decoder Training scheme that encourages the encoder to learn richer spatial representations by jointly modeling metric and relative depth abstractions. Fig. 1 shows the results from our Distil-E2D under challenging low-light conditions.

To our knowledge, this is the first approach to explicitly distill depth knowledge from the image domain into the event domain by leveraging naturally co-registered APS frames and co-calibrated external RGB cameras within real-world event datasets. Unlike synthetic simulation pipelines that suffer from event domain shift, our method operates entirely on real event data. This preserves the inherent dynamics and statistics of event streams , which result in models that are more robust and generalizable in practice. These innovations position our Distil-E2D as a principled framework for bridging the supervision gap in event-based depth learning with real-world robustness and practical scalability. Our contributions are summarized as follows:

- We propose **Distil-E2D**, the first framework to distill dense depth priors from image-based foundational models into the event domain using co-registered APS or RGB frames.

- We introduce a **Confidence-guided Calibrated Depth Loss (CCDL)** to reconcile synthetic and real depth signals through learned **Nonlinear Depth Calibration (NDC)** and **Alignment-aware Confidence Estimation (ACE)**.

- We design a novel architecture that incorporates a **Context Transformer (CT)** to improve temporal modeling and a **Dual-Decoder Training (DDT)** strategy to improve spatial representation learning.

- We demonstrate **state-of-the-art results** across real-world benchmarks for both event-only and event+APS modalities, which validate the robustness and generalizability of our method.

## 2 Related Work

### 2.1 Event-based Monocular Depth Estimation

Early approaches to event-based monocular depth estimation relied on geometric methods [24–27], which estimated depth by optimizing constraints from structure-from-motion or visual-inertial odometry. Despite being physically grounded, these methods require auxiliary data such as accurate camera poses or IMU readings and are sensitive to noise. To overcome these limitations, later works explored supervised learning approaches [28, 8] which convert asynchronous event streams into dense spatiotemporal representations for convolutional networks to regress depth. To compensate for the sparsity of event data, several methods incorporate additional input from APS or RGB sensors [9, 12, 29, 30] and fuse features from multiple modalities to improve prediction. More recently, Transformer-based [31] architectures [11, 12, 29] have been introduced for modeling long-range spatio-temporal dependencies via self- and cross-attention. However, the generalization ability of these learning-based models are limited due to their reliance on small datasets with sparse LiDAR supervision.

To mitigate data scarcity, some works leverage synthetic datasets by simulating event streams and corresponding depth maps from virtual environments [8, 9]. Although this enables large-scale training, such datasets suffer from two key domain gaps: 1) the event generation models often lack physical realism, and 2) the simulated scene structures fail to capture the complexity of real-world environments. These consequently lead to degraded downstream performance. An alternative is self-supervised learning [30], which uses estimated egomotion and photometric consistency to derive pseudo-depth labels. However, inaccuracies in pose estimation and image warping often result in noisy supervision that fails to capture fine-grained depth details. In contrast to synthetic or self-supervised approaches, our method retains the use of real-world event data while distilling dense and semantically rich supervision from powerful image-based depth priors to guide fine-grained depth estimation in the event domain.

### 2.2 Foundational Depth Models

Recent progress in foundational models has led to the development of highly generalizable monocular depth estimators [18, 19, 17, 15, 32, 13, 14] trained on diverse and large-scale datasets spanning indoor, outdoor, and synthetic scenes. These models depart from task-specific architectures by embracing scalable transformer or hybrid encoder-decoder designs capable of handling a broad range of environments without needing fine-tuning. Notable examples include DPT [18] and MiDaS [19], which paved the way for using large pretraining datasets and cross-domain learning to enable zero-shot generalization.

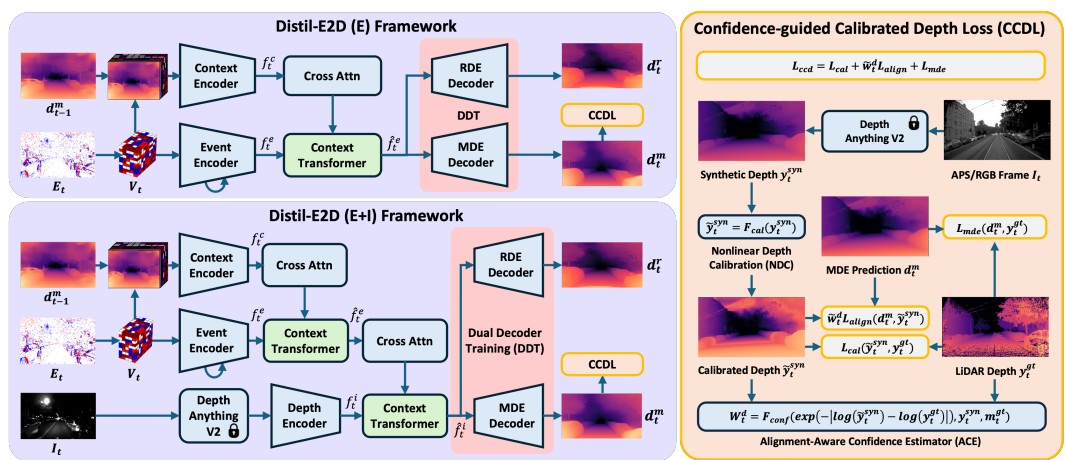

Figure 2: (Left) Distil-E2D architecture supporting event-only (E) and event+image (E+I) inputs. (Right) Overview of the Confidence-guided Calibrated Depth Loss (CCDL).

Depth Anything V2 [14] builds on the foundation of Depth Anything [13], which leverages pretrained vision models such as DINO [33] and Segment Anything [34] within a modular encoder-decoder architecture trained on a curated mix of high-resolution synthetic and real-world depth datasets. Although the original model demonstrated strong generalization across diverse domains, Depth Anything V2 significantly advances the state-of-the-art by addressing the limitations of real ground truth depth that is often sparse, noisy, or coarse. It proposes a novel strategy of generating high-quality pseudolabels from large-scale unlabeled real-world images to augment training. These synthetic depths retain fine-grained geometry and realistic scene structure, offering more detailed and semantically rich supervision than conventional real labels. With the use of pseudolabeled real images, Depth Anything V2 enhances depth fidelity, semantic precision, and generalization to set a new benchmark in monocular depth estimation.

This approach inspires our own strategy of using calibrated, confidence-weighted synthetic depths derived from real-world event data to augment sparse supervision and bridge the domain gap in event-based depth estimation. By aligning synthetic depths with real distributions and emphasizing reliable labels, we create high-quality supervision signals that enable fine-grained semantically-aware depth estimation from events.

## 3 Our Method

**Formulation.** Event cameras output asynchronous streams of events at the pixel level, capturing changes in log intensity with high temporal precision. Each pixel independently monitors log-intensity signal $L(\mathbf{u}, t)$, at pixel location $\mathbf{u}$ and time $t$. An event $e_i = (\mathbf{u}_i, t_i, p_i)$ is generated whenever the change in log-intensity exceeds a contrast threshold $C_{\text{thres}}$:

$$\Delta L(\mathbf{u}_i, t_i) = L(\mathbf{u}_i, t_i) - L(\mathbf{u}_i, t_i - \delta t), \quad |\Delta L(\mathbf{u}_i, t_i)| \geq C_{\text{thres}}, \tag{1}$$

with the polarity $p_i \in \{-1, +1\}$ indicating an increase or decrease in brightness. We represent a window of events as:

$$E = \{e_i\}_{i=1}^{N}, \tag{2}$$

where $N$ is the number of events in the interval. Given an event window $E_{[t-T,t]}$, the aim in EMDE is to predict a dense metric depth map

$$d_t^m = \mathcal{F}_{\text{e2d}}(E_{[t-T,t]}) \in \mathbb{R}^{H \times W}, \tag{3}$$

where $\mathcal{F}_{\text{e2d}}$ is the learned event-to-depth model and $H \times W$ is the image resolution.

**Event Representation.** Following prior works [8, 9, 35, 36], we convert the asynchronous event stream into a grid-based voxel representation. Given a stream of events $E = \{(\mathbf{u}_i, t_i, p_i)\}_{i=1}^{N}$ within a fixed temporal window $\Delta T$, we divide this window into $B$ temporal bins to construct a voxel

grid $V \in \mathbb{R}^{H \times W \times B}$, where $H$ and $W$ are the spatial dimensions of the sensor, and $B$ denotes the temporal resolution. Each event is first temporally normalized to the range $[0, B-1]$ using:

$$t_i^* = \frac{(B-1)(t_i - T_0)}{\Delta T}, \tag{4}$$

where $T_0$ is the start time of the event slice. We then perform linear interpolation to distribute the polarity of each event $p_i$ across its nearest temporal bins. The voxel value at $(\mathbf{u}, t)$ is accumulated as:

$$V(\mathbf{u}, t) = \sum_i p_i \cdot \max(0, 1 - |t - t_i^*|)\delta(\mathbf{u} - \mathbf{u}_i), \tag{5}$$

which ensures that contributions are smoothly distributed over time. This results in a dense spatiotemporal tensor that encodes both the spatial layout and fine-grained motion cues of the event stream for effective learning of dynamic structure in monocular depth estimation.

## 3.1 Dense Depth Prior Distillation

To enhance supervision in EMDE, we distill dense depth priors from pretrained image-based foundational depth models. Specifically, we use **DepthAnythingV2** [14], a state-of-the-art monocular depth predictor trained on large-scale RGB-depth datasets to generate *synthetic depth pseudolabels* from image frames temporally aligned with events. These frames may be (i) APS images captured by hybrid event cameras (e.g., DAVIS240C) or (ii) RGB frames from synchronized external cameras. The resulting synthetic depths serve as auxiliary supervision signals during training. Fig.3 illustrates the supervision signals. Compared to the sparse and artifact-laden LiDAR depth maps, the synthetic pseudolabels are significantly denser, smoother, and more spatially coherent, especially around

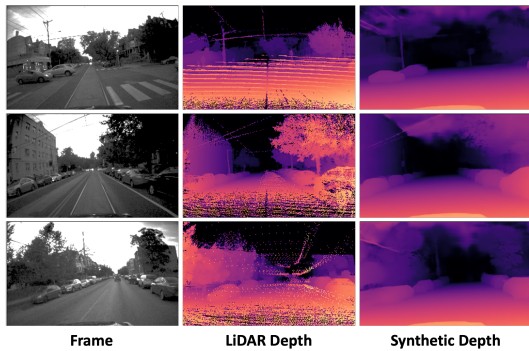

**Frame**  **LiDAR Depth**  **Synthetic Depth**

Figure 3: LiDAR vs. synthetic depth. LiDAR is sparse, incomplete, and exhibits artifacts, while synthetic depths are dense and consistent.

object boundaries. This enables more effective spatial learning from sparse event inputs. To maximize label quality, we apply DepthAnythingV2 primarily to well-lit daytime scenes. Formally, for an image frame $I_t$ at time $t$, we define the synthetic pseudolabel as:

$$y_t^{\text{syn}} = \mathcal{F}_{\text{da}}(I_t), \quad y_t^{\text{syn}} \in \mathbb{R}^{H \times W}, \tag{6}$$

where $\mathcal{F}_{\text{da}}$ denotes the pretrained DepthAnythingV2 model. The resulting depth map $y_t^{\text{syn}}$ is then used alongside sparse LiDAR ground truth $y_t^{\text{gt}}$ to supervise the event-based depth prediction network. This cross-modal distillation transfers rich structural priors from the image domain into the event domain to improve depth estimation in regions where LiDAR is missing or unreliable.

## 3.2 Confidence-guided Calibrated Depth Loss (CCDL)

The Confidence-guided Calibrated Depth Loss (CCDL) aims to metrically align synthetic depth pseudolabels and emphasize reliable spatial regions to provide useful synthetic supervision. Illustrated in Fig. 2, the CCDL comprises two subcomponents: 1) Nonlinear Depth Calibration (NDC) to metrically align synthetic depths to LiDAR depths, and 2) Alignment-aware Confidence Estimation (ACE) to emphasize spatial supervision based on mapped depth agreement.

**Nonlinear Depth Calibration (NDC).** The dense predictions from DepthAnythingV2 are *relative* depth estimates (RDE) that lack metric scale. In contrast, EMDE is supervised using *metric* depths derived from LiDAR. Consequently, synthetic depths cannot be directly used as pseudolabels without appropriate calibration. Existing methods [8, 9, 11, 12, 18, 19, 13] commonly rely on the *scale-invariant loss* $\mathcal{L}_{\text{si}}$ [8] to address this mismatch, which assume a global scale and shift discrepancy between relative and metric spaces. However, this scale-shift assumption is often over-simplified. In practice, relative depths exhibit *nonlinear scale compression*, especially in scenes with large depth

variation. To address this issue, we propose learning a nonlinear calibration function using a small MLP $\mathcal{F}_{\text{cal}}$, which maps synthetic log-depths $\log y_t^{\text{syn}}$ to calibrated log-depths $\log \tilde{y}_t^{\text{syn}}$:

$$\log \tilde{y}_t^{\text{syn}} = \mathcal{F}_{\text{cal}}(\log y_t^{\text{syn}}), \quad \mathcal{L}_{\text{cal}} = \|\log \tilde{y}_t^{\text{syn}} - \log y_t^{\text{gt}}\|_2^2, \quad (7)$$

where $\mathcal{L}_{\text{cal}}$ is the loss function. This allows us to correct systematic nonlinear distortions in the synthetic depths for better alignment to real-world metric scales. Using a logarithmic depth representation improves numerical stability and emphasizes relative depth relationships, which is important for aligning synthetic and LiDAR depths. Driving scenes span a large depth range, from near objects under 2 m to distant backgrounds tens of meters away. The logarithmic transform compresses this range into a bounded interval (approximately 0–1 in our normalized setup), allowing both near and far regions to contribute comparably to the loss.

The calibrated depths $\tilde{y}_t^{\text{syn}}$ are subsequently used to supervise the metric depth predictions $d_t^m$ from our Distil-E2D network using an alignment loss:

$$\mathcal{L}_{\text{align}}(d_t^m, \tilde{y}_t^{\text{syn}}) = \mathcal{L}_{\text{si}}(d_t^m, \tilde{y}_t^{\text{syn}}) + \mathcal{L}_{\text{gm}}(d_t^m, \tilde{y}_t^{\text{syn}}), \quad (8)$$

where $\mathcal{L}_{\text{gm}}$ denotes the multi-scale gradient matching loss from [8]. Concurrently, metric predictions also receive supervision from sparse LiDAR depths:

$$\mathcal{L}_{\text{mde}}(d_t^m, y_t^{\text{gt}}) = \mathcal{L}_{\text{si}}(d_t^m, y_t^{\text{gt}}) + \mathcal{L}_{\text{gm}}(d_t^m, y_t^{\text{gt}}). \quad (9)$$

The use of calibrated synthetic depths provides dense and semantically rich supervision that complements sparse LiDAR signals. This encourages better generalization by exposing the model to a broader range of depth cues during training. Dense supervision helps the network learn fine-grained spatial structures, such as object boundaries and scene layout, which are difficult to infer from sparse labels alone.

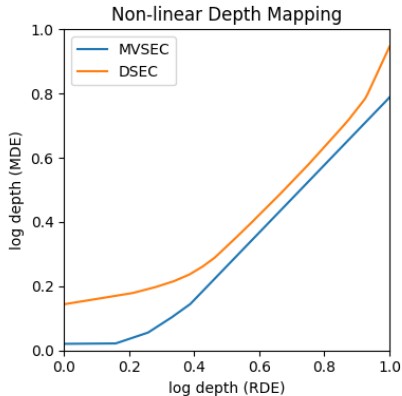

Figure 4: Learned nonlinear maps for relative-to-metric depth alignment.

**Alignment-Aware Confidence Estimation (ACE).** However, nonlinear calibration may leave residual misalignments that make some synthetic depth regions less reliable. To address this issue, we introduce an Alignment-Aware Confidence Estimator (ACE) that assigns per-pixel reliability scores given a calibrated depth map $\tilde{y}_t^{\text{syn}}$ and a sparse LiDAR depth map $y_t^{\text{gt}}$:

$$w_t^s = \exp(-|\log \tilde{y}_t^{\text{syn}} - \log y_t^{\text{gt}}|), \quad w_t^s \in \mathbb{R}^{H \times W}. \quad (10)$$

The ACE formulation ensures that: 1) well-aligned regions receive higher weights, and 2) a fixed absolute error at larger ground-truth depth results in smaller penalization. Due to the sparsity of $y_t^{\text{gt}}$, the initial confidence map $w_t^s$ is also sparse. Therefore, we train a lightweight CNN $\mathcal{F}_{\text{conf}}$ to propagate the sparse confidences to produce a dense confidence map $w_t^d$:

$$w_t^d = \mathcal{F}_{\text{conf}}(w_t^s, y_t^{\text{syn}}, m_t^{\text{gt}}), \quad (11)$$

where $m_t^{\text{gt}}$ is the binary validity mask indicating valid depth locations in $y_t^{\text{gt}}$. The network leverages the synthetic

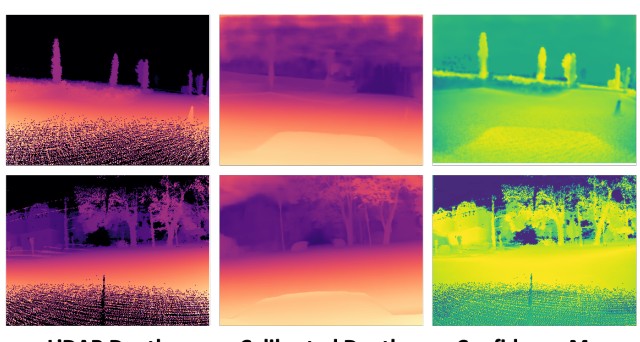

**LiDAR Depth**   **Calibrated Depth**   **Confidence Map**

Figure 5: ACE confidence maps at epoch 1 (top row) and epoch 200 (bottom row).

depth map to provide spatial context and uses $m_t^{\text{gt}}$ to provide LiDAR location cues. The resulting dense confidence map $w_t^d$ is then used to weigh $\mathcal{L}_{\text{align}}$, emphasizing supervision in spatially reliable regions.

To prevent collapse of $w_t^d$, we incorporate two stabilizing mechanisms: **base clamping** and a **sparse confidence regularizer**. Specifically, we reparametrize:

$$\tilde{w}_t^d = \alpha + (1 - \alpha)w_t^d, \tag{12}$$

with $\alpha = 0.1$. This enforces $\tilde{w}_t^d \in [0.1, 1]$ to avoid vanishing gradients. Additionally, we apply an L1 regularization loss:

$$\mathcal{L}_{\text{conf}} = \|\tilde{w}_t^d - \hat{w}_t^s\|_1, \tag{13}$$

where $\hat{w}_t^s$ is the min-max normalized sparse confidence map from section 3.2. This regularizer encourages consistency with high-confidence regions and promotes non-trivial and spatially adaptive weighting.

Figure 5 illustrates the evolution of the ACE confidence maps from epoch 1 (top) to epoch 200 (bottom). Initially, the confidence weights are poorly defined due to the lack of reliable correspondence between synthetic and ground-truth depths. As training progresses, the confidence maps converge toward coherent spatial patterns that align closely with scene geometry, highlighting regions where depth alignment is more consistent. This progression shows that $\mathcal{F}_{\text{conf}}$ learns to propagate confidence from sparse LiDAR locations to structurally meaningful areas, enabling adaptive weighting of $\mathcal{L}_{\text{align}}$. Consequently, the network focuses training on geometrically reliable regions, leading to more stable and accurate depth learning.

Combining NDC and ACE, we define the final training objective as CCDL:

$$\mathcal{L}_{\text{ccd}} = \mathcal{L}_{\text{cal}} + \tilde{w}_t^d \cdot \mathcal{L}_{\text{align}}(d_t^m, \tilde{y}^{\text{syn}}) + \mathcal{L}_{\text{mde}}(d_t^m, y^{\text{gt}}). \tag{14}$$

This composite loss combines dense calibrated synthetic priors, sparse metric supervision, and adaptive weighting to effectively guide the learning of EMDE.

### 3.3 Network Architecture

As illustrated in Fig. 2 (left), we introduce two model variants: Distil-E2D(E) (left, top) for event-only input, and Distil-E2D(E+I) (left, bottom) for configurations including co-recorded APS or RGB frames. Distil-E2D(E) comprises two main branches: 1) an **event branch** that employs a recurrent convolutional encoder to process event voxel sequences, and 2) a **context branch** that encodes depth prediction of the previous timestep along with the current event voxel. 3) An **image branch** is introduced in Distil-E2D(E+I), which uses DepthAnythingV2 [14] as the frame encoder. Both models incorporate a Context Transformer for temporal integration and a Dual-Decoder Training regime to improve the quality of the encoder representation. The encoders in the context, event and image branches produce feature tensors $f_t^c$, $f_t^e$, and $f_t^i$, respectively.

**Context Transformer (CT).** As highlighted in green in Fig.2, we introduce the CT modules in the bottleneck region of our network. The CT module connects the **event branch** to the **context branch**, and also the **image branch** to the **event branch** in Distil-E2D(E+I). To incorporate past context into current predictions, we refine the event features $f_t^e$ by attending to the context features $f_t^c$ via cross-attention:

$$\text{CrossAttention}(Q, K, V) = \text{softmax}\left(\frac{QK^\top}{\sqrt{d_k}}\right)V, \quad Q = f_t^e W^Q, \quad K = f_t^c W^K, \quad V = f_t^c W^V, \tag{15}$$

where $f_t^e$ serves as the query and $f_t^c$ supplies the keys and values. Multi-head cross-attention is performed within a Transformer module placed at the bottleneck of the event branch. The refined event features are denoted by:

$$\hat{f}_t^e = \mathcal{F}_{\text{evtr}}(f_t^e, f_t^c), \quad \hat{f}_t^e \in \mathbb{R}^{\frac{H}{16} \times \frac{W}{16}}, \tag{16}$$

where $\mathcal{F}_{\text{evtr}}$ denotes the event-branch transformer module. In Distil-E2D(E+I), the image branch updates $f_t^i$ with refined event features $\hat{f}_t^e$:

$$\hat{f}_t^i = \mathcal{F}_{\text{imtr}}(f_t^i, \hat{f}_t^e), \quad \hat{f}_t^i \in \mathbb{R}^{\frac{H}{16} \times \frac{W}{16}}, \tag{17}$$

where $\mathcal{F}_{\text{imtr}}$ is the image-branch transformer module. We use two Transformer blocks per branch with a multi-head self-attention layer followed by a multi-head cross-attention layer, respectively. Attention is applied in a low-resolution latent space for per-pixel reasoning. Positional encodings are learned for each latent pixel to preserve spatial consistency. This architecture allows efficient extraction of relevant temporal and semantic context for more accurate and consistent depth predictions.

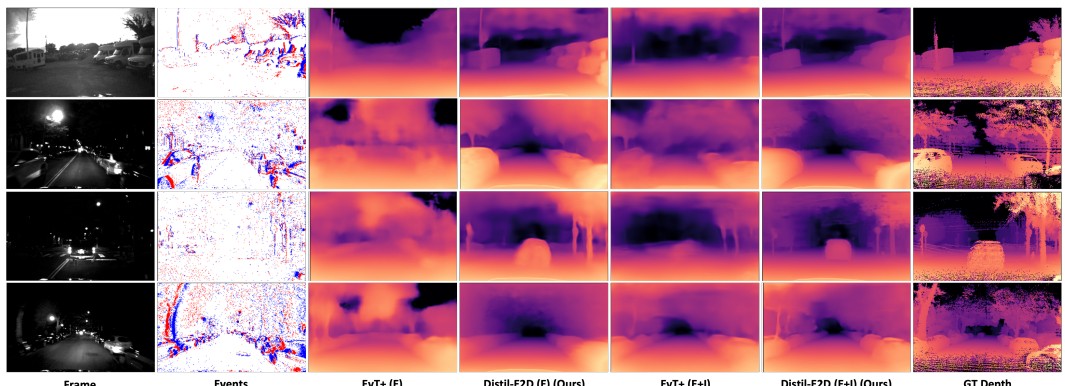

| Frame | Events | EvT+ (E) | Distil-E2D (E) (Ours) | EvT+ (E+I) | Distil-E2D (E+I) (Ours) | GT Depth |

Figure 6: (Left) Qualitative results on MVSEC Dataset.

Table 1: Quantitative results on MVSEC Dataset.

|  | day1 | | | night1 | | | night2 | | | night3 | | | |
|---|---|---|---|---|---|---|---|---|---|---|---|---|---|
|  | 10m ↓ | 20m ↓ | 30m ↓ | 10m ↓ | 20m ↓ | 30m ↓ | 10m ↓ | 20m ↓ | 30m ↓ | 10m ↓ | 20m ↓ | 30m ↓ | runtime(ms) |
| **Event-only (E)** | | | | | | | | | | | | | |
| ULODE[28] | 2.72 | 3.84 | 4.40 | 3.13 | 4.02 | 4.89 | 2.19 | 3.15 | 3.92 | 2.86 | 4.46 | 5.05 | 25.0 |
| MDDE+[8] | 1.85 | 2.64 | 3.13 | 3.38 | 3.82 | 4.46 | 1.67 | 2.63 | 3.58 | 1.42 | 2.33 | 3.18 | 8.0 |
| EMD[30] | 1.40 | 2.07 | 2.65 | 2.18 | 2.70 | 3.64 | 2.06 | 2.76 | 3.42 | 2.09 | 2.82 | 3.52 | **5.3** |
| ERE[11] | 1.44 | 2.23 | 2.75 | 1.85 | 2.45 | 3.24 | 1.66 | 2.53 | 3.15 | 1.49 | 2.37 | 2.97 | 35.0 |
| EvT+ (E)[12] | 1.31 | 1.92 | 2.32 | 1.54 | 2.31 | 2.96 | 1.47 | 2.22 | 2.92 | 1.36 | 2.13 | 2.84 | 35.0 |
| Ours (E) | **1.14** | **1.56** | **1.81** | **1.41** | **2.16** | **2.75** | **1.33** | **2.06** | **2.70** | **1.13** | **2.04** | **2.79** | 10.5 |
| **Event+Image (E+I)** | | | | | | | | | | | | | |
| RAMNet[9] | 1.39 | 2.17 | 2.76 | 2.50 | 3.19 | 3.82 | 1.21 | 2.31 | 3.28 | 1.01 | 2.34 | 3.43 | **6.2** |
| EvT+ (E+I)[12] | 1.24 | 1.91 | 2.36 | 1.45 | 2.10 | 2.88 | 1.48 | 2.13 | 2.90 | 1.38 | 2.03 | 2.77 | 52.0 |
| Ours (E+I) | **1.04** | **1.44** | **1.75** | **1.35** | **2.02** | **2.79** | **1.20** | **2.04** | **2.62** | **0.99** | **1.96** | **2.71** | 47.5 |

**Dual-Decoder Training (DDT).** As highlighted in pink in Fig. 2, we introduce two decoders: the **MDE Decoder** for metric depth estimation and the **RDE Decoder** for relative depth estimation to enrich the feature representations of the encoder. The MDE Decoder predicts $d_t^m$ and is supervised using CCDL (section 3.2), and the RDE Decoder predicts $d_t^r$ and is trained with uncalibrated synthetic depths $y^{\text{syn}}$ via:

$$\mathcal{L}_{\text{rde}}(d_t^r, y^{\text{syn}}) = \mathcal{L}_{\text{si}}(d_t^r, y^{\text{syn}}) + \mathcal{L}_{\text{gm}}(d_t^r, y^{\text{syn}}).$$

The RDE Decoder used only during training provides dense supervision to accelerate convergence and guide the encoder to focus on fine-grained scene details: object boundaries and edge consistency, which are often missed with sparse LiDAR supervision. Using separate MDE and RDE Decoders also prevents gradient interference and allows each to specialize in its respective depth abstraction while jointly regularizing the encoder towards a robust and generalizable feature space.

## 4 Experiments

### 4.1 Experimental Setup

We evaluate our proposed **Distil-E2D** framework under two settings: (E) using event data only, and (E+I) using both event and image (APS or RGB). We benchmark Distil-E2D (E) against five prior event-only methods: ULODE[28], MDDE+[8], EMD[30], ERE[11] and EvT+ (E)[12] . For the (E+I) configuration, we compare against multi-modal methods RAMNet[19] and EvT+ (E+I)[12]. We evaluate against two real-world driving datasets:

**1) MVSEC.** This dataset [2] is a standard benchmark for event-based depth estimation, comprising synchronized events, grayscale APS images, and LiDAR depth from outdoor driving sequences recorded with a DAVIS 240C ($260 \times 346$). Following prior work, we use the `outdoor_day_2` sequence for training and evaluate on the remaining four sequences.

**2) DSEC.** This dataset [3] provides high-resolution stereo event and RGB data ($480 \times 640$ events, $1080 \times 1440$ RGB frames) along with LiDAR depths across 41 driving sequences under diverse

Table 2: Ablations on MVSEC Dataset.

| SYN | NDC | ACE | CT | DDT | day1 | | | night1 | | | night2 | | | night3 | | |
|---|---|---|---|---|---|---|---|---|---|---|---|---|---|---|---|---|
| | | | | | 10m ↓ | 20m ↓ | 30m ↓ | 10m ↓ | 20m ↓ | 30m ↓ | 10m ↓ | 20m ↓ | 30m ↓ | 10m ↓ | 20m ↓ | 30m ↓ |
| ✓ | ✗ | ✗ | ✗ | ✗ | 2.89 | 4.01 | 3.91 | 3.53 | 4.01 | 5.52 | 2.63 | 3.56 | 4.01 | 2.29 | 3.12 | 4.11 |
| ✗ | ✗ | ✗ | ✗ | ✗ | 2.63 | 3.52 | 3.68 | 5.96 | 4.63 | 5.11 | 2.91 | 3.05 | 3.96 | 2.62 | 2.73 | 3.66 |
| ✓ | ✓ | ✗ | ✗ | ✗ | 1.47 | 1.95 | 2.41 | 1.74 | 2.45 | 3.22 | 1.60 | 2.43 | 3.33 | 1.35 | 2.23 | 2.97 |
| ✓ | ✓ | ✓ | ✗ | ✗ | 1.35 | 1.76 | 2.01 | 1.53 | 2.32 | 3.04 | 1.47 | 2.17 | 3.01 | 1.21 | 2.14 | 2.82 |
| ✓ | ✓ | ✓ | ✓ | ✗ | 1.19 | 1.59 | 1.86 | 1.44 | 2.21 | 2.79 | 1.36 | 2.10 | 2.78 | 1.18 | 2.06 | 2.81 |
| ✓ | ✓ | ✓ | ✓ | ✓ | **1.14** | **1.56** | **1.81** | **1.41** | **2.16** | **2.75** | **1.33** | **2.06** | **2.70** | **1.13** | **2.04** | **2.79** |

lighting conditions. We use 28 sequences for training and 13 for testing. Since [30] is the only work reporting on DSEC, we compare directly with their results.

## 4.2 Quantitative Results

Table 1 reports the quantitative results on MVSEC following the standard evaluation protocol of the prior works with average depth errors at cutoff distances of 10m, 20m, and 30m. Our Distil-E2D consistently outperforms all supervised baselines. This demonstrates that the use of dense and high-quality calibrated pseudolabels distilled from foundation depth models yields greater performance gains than solely relying on sparse LiDAR supervision.

Distil-E2D (E) and (E+I) exceed the performance of MDDE+[8] and RAMNet[9], both of which utilize large-scale simulated event data for training. This highlights the ability of our framework to bridge the synthetic-to-real gap more effectively by combining real-world event data with calibrated confidence-weighted synthetic pseudola-

Table 3: Results on DSEC Dataset.

| | 10m ↓ | 20m ↓ | 30m ↓ |
|---|---|---|---|
| EMD [30] | 0.96 | 1.55 | 2.21 |
| Ours (E) | **0.64** | **1.11** | **1.61** |

bels. Comparing with self-supervised methods, Distil-E2D (E) surpasses EMD [30]. This shows the superiority of depth priors distilled from image foundation models over photometric-consistency-based pseudo-supervision.

Distil-E2D also outperforms [30] on DSEC, which shows the scalability of our method to higher resolution event data and its compatibility with both APS and RGB modalities. As shown in Figure 7, higher-resolution image frames yield finer and sharper depth estimates which further validates our approach of a principled distillation of dense depth priors from foundational models.

## 4.3 Qualitative Results

Qualitative results are shown in Fig. 6 for MVSEC and Fig. 7 for DSEC. On MVSEC, Distil-E2D produces sharper object boundaries such as cars, trees, and lampposts compared to the next best method EvT+ [12] which remains visibly blurred. For example, in row 1, our Distil-E2D clearly delineates the parked vehicles. On the other hand, methods trained only with sparse LiDAR [11, 12] yield oversmoothed predictions. This is because EMDE inherently requires semantic understanding and boundary precision, which are better captured by dense and semantically rich pseudolabels generated from foundational models than by sparse LiDAR supervision. As a result, our Distil-E2D more accurately segments and assigns depth to fine scene structures.

Similar patterns are observed in the DSEC results (Fig.7). Although comparisons with EMD[30] remain quantitative due to unavailable code, our Distil-E2D produces visually accurate predictions. For example, in Fig. 7 (row 3, column 4), LiDAR fails to capture the full extent of the safety bollard and road barrier. In contrast, our Distil-E2D reconstructs them accurately. This underscores the value of foundation model–based pseudo-label supervision for EMDE.

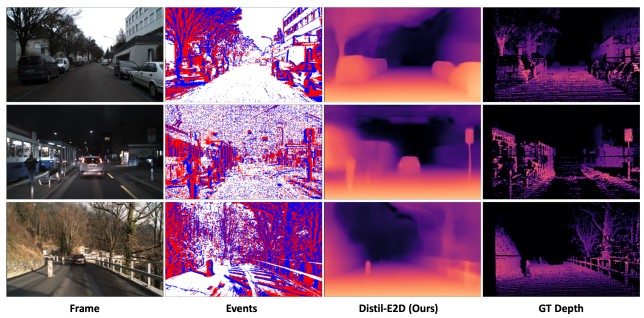

Figure 7: (Left) Qualitative results on DSEC Dataset.

### 4.4  Ablation Study

We present ablations in Tab. 2 to quantify the contribution of each core component in Distil-E2D:

**NDC.** Training on synthetic depth without NDC (row 1) causes the largest drop in performance, underperforming even the LiDAR-only baseline (row 2). With NDC (row 3) depth accuracy improves significantly. This validates the role of NDC in correcting scale and distribution mismatches between synthetic pseudolabels and the LiDAR ground truth. The result also confirms that scale-invariant losses alone are insufficient to bridge the nonlinear relative-metric scale gap, and that explicit nonlinear calibration is necessary for effectively leveraging dense synthetic supervision.

**ACE.** Introducing ACE (row 4) yields further gains, confirming its effectiveness in weighting supervision towards more reliable regions. This mitigates the impact of misaligned or noisy pseudolabels, allowing the model to prioritize cleaner training signals.

**CT.** Adding CT (row 5) leads to additional improvements, validating its ability to incorporate temporal context. By attending to prior predictions, CT enhances performance in frames with sparse events or ambiguous geometry.

**DDT.** Incorporating DDT (row 6) also improves performance. This shows that joint supervision on both metric and relative depth encourages the encoder to learn richer and more generalizable features with better semantic structure and spatial precision.

Overall, these results confirm that each component contributes uniquely to model performance. Together, they enable Distil-E2D to achieve strong EMDE results through more effective use of synthetic supervision, calibrated integration, and temporal reasoning.

## 5  Conclusion

We present **Distil-E2D**, a framework for event-based monocular depth estimation that distills dense priors from pretrained image-based depth models. By leveraging synthetic pseudolabels, our method surpasses the limitations of sparse LiDAR supervision. Our key contributions include Nonlinear Depth Calibration (NDC) for scale alignment, Alignment-aware Confidence Estimation (ACE) for robust supervision, a Context Transformer (CT) for temporal modeling, and Dual-Decoder Training (DDT) for feature generalization. These components collectively enable Distil-E2D to achieve state-of-the-art performance. Our results highlight the potential of foundation model distillation to advance generalizable depth perception in the event domain.

### Acknowledgments

This research / project is supported by the National Research Foundation, Singapore, under its NRF-Investigatorship Programme (Award ID. NRF-NRFI09-0008) and AI Singapore Programme (AISG Award No: AISG2-PhD/2021-08-012), and the Tier 1 grant T1-251RES2305 from the Singapore Ministry of Education.

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

# A   Supplementary Material

## A.1   Limitations

Although Distil-E2D achieves state-of-the-art results across multiple benchmarks, it is subject to several limitations. First, the framework is dependent on the availability and quality of dense, calibrated pseudolabels derived from image-based foundation models. Inaccurate or biased depth estimates from these models, especially in low-light environments or textureless regions, can degrade the effectiveness of the training signal. Second, although Distil-E2D improves the quality of supervision through dense synthetic depths, the quantity and diversity of event-based training data remain limited. This constrains the model's ability to generalize to complex, real-world driving conditions with varied lighting, geometry, and motion patterns.

Future work could mitigate these issues by leveraging ensembles of depth foundation models to reduce pseudolabel bias, integrating online refinement techniques for self-correction during training, and employing video-to-event (vid2e) generation strategies to synthesize realistic event data from large-scale monocular video datasets. Distil-E2D enables a promising direction: By coupling with video-to-event (vid2e) conversion strategies, it can transform large-scale monocular video datasets into synthetic event–depth datasets, significantly reducing the need for laborious data collection and opening new avenues for scalable event-based depth learning.

## A.2   Implementation Details

**Data Preparation.**   Each depth map is paired with events from the preceding $50$ms window. To address frame rate differences between the LiDAR and RGB/APS modalities, we match each depth map to the nearest available image frame within a $\pm 50$ms margin. If no valid match is found, we exclude the image frame from synthetic depth generation and loss computation. These images are passed through DepthAnythingV2-Large to generate dense synthetic pseudolabels. Event data is discretized into spatiotemporal voxel grids with $B = 5$ bins for MVSEC and $B = 15$ bins for DSEC to accommodate varying event densities due to sensor resolution. During training, we apply horizontal flipping to events, images, and depth maps for data augmentation. All code, pre-trained weights, and training configurations will be released to facilitate reproducibility.

**Encoder Architecture.**   We use DepthAnythingv2-L for the frame encoder in the E+I setting. The architecture of our event encoder, context encoder, and RDE/MDE decoder is as follows:

**Event Encoder (Recurrent):**

- **Input Head:** Conv2d(5, 32, kernel size 5, stride 1, padding 2)
- **Encoder 0:** Conv2d(32, 64, kernel size 5, stride 2) + ConvLSTM(128, 256, kernel size 3)
- **Encoder 1:** Conv2d(64, 128, kernel size 5, stride 2) + ConvLSTM(256, 512, kernel size 3)
- **Encoder 2:** Conv2d(128, 256, kernel size 5, stride 2) + ConvLSTM(512, 1024, kernel size 3)
- **Encoder 3:** Conv2d(256, 512, kernel size 5, stride 2) + ConvLSTM(1024, 2048, kernel size 3)

**Context Encoder (Feedforward):**

- **Input Head:** Conv2d(6, 32, kernel size 5, stride 1, padding 2)
- **Encoder 0:** Conv2d(32, 64, kernel size 5, stride 2)
- **Encoder 1:** Conv2d(64, 128, kernel size 5, stride 2)
- **Encoder 2:** Conv2d(128, 256, kernel size 5, stride 2)
- **Encoder 3:** Conv2d(256, 512, kernel size 5, stride 2)

**Decoder (Shared for RDE and MDE Heads):**

- **Decoder 0:** Upsample + Conv2d(512, 256, kernel size 5, stride 1)
- **Decoder 1:** Upsample + Conv2d(256, 128, kernel size 5, stride 1)

- **Decoder 2:** Upsample + Conv2d(128, 64, kernel size 5, stride 1)
- **Decoder 3:** Upsample + Conv2d(64, 32, kernel size 5, stride 1)
- **Output Head:** Conv2d(32, 1, kernel size 1)

Skip connections are applied between the event encoder and RDE/MDE decoders.

**Nonlinear Depth Calibrator (NDC).** The NDC module is a three-layer multilayer perceptron with a 1D input and output. It contains a single hidden layer of 64 units and learns a nonlinear mapping function that transforms relative synthetic depths to the metric scale of ground-truth LiDAR. This component plays a crucial role in reconciling distributional differences between synthetic and real depths.

**Alignment-aware Confidence Estimator (ACE).** The ACE module is a four-layer convolutional neural network with a 3-channel input (concatenation of the sparse confidence map, synthetic depth, and the sparse LiDAR depth mask), 32-channel hidden representation, and 1-channel confidence output. All convolutions use $3 \times 3$ filters with padding 1 to preserve spatial resolution. A final sigmoid activation ensures that confidence weights fall within $[0, 1]$, modulating the contribution of each pixel to the loss based on alignment quality.

**Training Configuration.** Distil-E2D is implemented using PyTorch 2.0 and trained with the AdamW optimizer at a learning rate of $1 \times 10^{-4}$. A OneCycleLR scheduler dynamically adjusts the learning rate. Training is conducted over 250 epochs with full precision and a batch size of 16, using sequences of 10 frames corresponding to a $500$ms event window.

## A.3 Analysis of Context-Induced Distortions under Accelerated Motion.

We analyze whether the context branch introduces distortions under acceleration on the MVSEC test set using event-only input. The test frames are segmented into low, medium, and high motion bins based on the average ground truth optical flow magnitude with thresholds defined by the 33rd and 66th percentiles. For each bin, we report the mean depth error at 30m.

| Motion Bin | Baseline (no context) | Ours (w/ context) | $\Delta$ |
|---|---|---|---|
| Low | 2.66 | **2.34** | 0.32 |
| Medium | 2.71 | **2.52** | 0.19 |
| High | 2.80 | **2.69** | 0.11 |

The results show that with the context branch, the accuracy decreases with increasing motion due to the growing misalignment between frames, which reduces the effectiveness of temporal fusion. Nonetheless, the context branch continues to provide valuable temporal cues and structural continuity, yielding improved accuracy over the baseline across all motion bins.

## A.4 Compute Resources

All experiments were performed on a single NVIDIA RTX A6000 GPU with 48 GB of memory. Generating pseudolabels from DepthAnythingV2-Large required 4-6 hours per dataset. Training Distil-E2D took approximately 72 hours for MVSEC and 96 hours for DSEC. We did not use distributed training or multi-GPU setups. The entire workflow is executable within compute budgets accessible to most academic institutions.

## A.5 License and Credit

We acknowledge the use of publicly available models and datasets, each used in compliance with their respective open-source licenses:

- **Depth Anything V2**: Apache 2.0 License
- **MVSEC Dataset**: Creative Commons BY 4.0 License
- **DSEC Dataset**: Creative Commons BY-SA 4.0 License

## A.6 Broader Societal Impact

This work advances event-based depth perception, enabling power-efficient, low-latency applications in domains such as autonomous robotics, AR/VR, and assistive systems. Although these capabilities are beneficial in fast-motion or low-light environments, they also introduce significant risks. Real-time depth inference using covert, low-data-rate event cameras may facilitate surveillance, unauthorized tracking, or military use. Furthermore, distillation of pretrained models can propagate existing biases or inadvertently expose sensitive information from uncurated datasets. These risks highlight the importance of transparency in dataset provenance, ethical deployment practices, and model auditing. We advocate for interdisciplinary oversight, fairness evaluations, and responsible use, especially in socially sensitive or surveillance-adjacent applications.

