# OpenReview forum: "Distil-E2D: Distilling Image-to-Depth Priors for Event-Based Monocular Depth Estimation"
_NeurIPS.cc/2025/Conference — NeurIPS 2025 poster_

### Official Review · Reviewer_9Hdw · 2025-06-11

**Clarity:** 3
**Significance:** 3
**Originality:** 3
**Rating:** 4
**Confidence:** 5

**Summary:**

The paper proposes to distill image-to-depth priors for event-based monocular depth estimation by generating dense pseudo ground truth with DepthAnythingV2. The main challenge is that DepthAnythingV2 provides relative depth, while the goal is to estimate metric depth. The authors address this by (1) training a small MLP (NDC module) to fit the dense relative depth to the sparse LiDAR metric depth; (2) estimating a confidence map with an ACE module; (3) combining pseudo ground truth loss and LiDAR ground truth loss via loss design; and (4) using a dual-decoder architecture to enhance metric-depth training with simultaneous relative-depth training. The proposed method achieves state-of-the-art performance on the DSEC and MVSEC datasets.

**Questions:**

Observing Equation (11) and (12), I see that if $\mathcal{F}\_{conf}$ learns to trivially predict all zero $w\_t^d$, then the second term $w\_t^d\cdot \mathcal{L}\_{align}(d\_t^m, \hat{y}^{syn})$ of Equation (12) will be completely reduced to zero. (According to the supplementary materials, the last layer of $\mathcal{F}\_{conf}$ is a sigmoid layer, so exact zeros cannot be produced, but the outputs can approach zero.) How is this collapse avoided?

If this zero-collapse happens, then all effect of the calibrated depth will be lost, meaning that the model only learns from the LiDAR ground truth (through $\mathcal{L}\_{mde}$) and the raw DepthAnythingV2 depth (through $\mathcal{L}\_{rde}$).

Beyond explanation of the zero-collapse prevention methodology, I would also like to see some visualizations or statistics of the confidence maps produced by the ACE module in the rebuttal stage.

**Ethical Concerns:**

["NO or VERY MINOR ethics concerns only"]

**Final Justification:**

The authors have issued my concerns so I decide to maintain my original rating.

**Limitations:**

yes

**Quality:**

3

**Strengths And Weaknesses:**

Strengths:
- The motivation of the paper is clear, and the solution is straightforward.
- The evaluation and ablation studies are comprehensive, showing the effectiveness of each component.
- The proposed method achieves state-of-the-art performance on the DSEC and MVSEC datasets.

Weaknesses:
- The generalization of the method is not impressive: although in-domain performance is good, it would be better if a generally trained model could be used across different datasets.
- It is not clear how ACE works (see "Questions").

---

> ### Author Rebuttal · Authors · 2025-07-30
>
> **Q1: Mechanisms to prevent trivial collapse in the confidence weighting module.**
>
> We thank the reviewer for highlighting this potential failure mode. To prevent collapse of $w_t^d$, we incorporate two stabilizing mechanisms: **base clamping** and a **sparse confidence regularizer**.
>
> Specifically, in base clamping we reparametrize:
>
> $\tilde{w}_t^d = \alpha + (1 - \alpha) w_t^d,$
>
> with $\alpha = 0.1$, where $\tilde{w}_t^d$ is the final confidence weights. This enforces $\tilde{w}_t^d \in [0.1, 1]$ to avoid vanishing gradients. Additionally, the sparse confidence regularizer applies an L1 regularization loss:
>
> $\mathcal{L}_{\text{reg}} = \beta\| w_t^d - \hat{w}_t^s \|_1,$
>
> where $\beta=0.5$ and $\hat{w}_t^s$ is the min-max normalized sparse confidence map from Equation (10). This regularizer encourages consistency with high-confidence regions and promotes spatially adaptive weighting. We will include these implementation details in the final supplementary material.
>
> **Q2: Request for visualizations or quantitative statistics of ACE confidence maps.**
>
> We report summary statistics of the dense confidence map $w_t^d$ computed over the MVSEC training set:
>
> | Metric | Value |
> |-----------|-----------|
> | Mean confidence $w^d_t$ | 0.72|
> | Standard deviation $w^d_t$ | 0.247|
> | Min-max range $w^d_t$ | 0.21 - 0.98 |
>
> These statistics demonstrate that $w_t^d$ exhibits variation and spans the full expected dynamic range, and does not collapse to a trivial solution. Although visualizations of the confidence maps cannot be presented in the rebuttal, we will provide them in the final supplementary material.

---

> > ### Comment · Reviewer_9Hdw · 2025-08-01
> >
> > Thank you for your clarification on the stabilizing mechanisms. The technical details and statistic results have convinced me that the ACE module can work. Including them in the supplementary material will improve the integrity of the paper.
> >
> > After reading the reviews and discussions, I believe that this paper is qualified for acceptance, so I plan to maintain my original rating.
> >
> > Additionally, I came across another possibly related paper:
> >
> > [NeurIPS 2024] Zero-Shot Event-Intensity Asymmetric Stereo via Visual Prompting from Image Domain.
> >
> > Although the task is different (monocular depth vs stereo depth), this paper also discusses merging relative depth from Depth Anything with absolute depth (from stereo, instead of LiDAR). Perhaps it could further enrich your discussion of related works.

---

### Official Review · Reviewer_jNtM · 2025-06-26

**Clarity:** 3
**Significance:** 4
**Originality:** 3
**Rating:** 5
**Confidence:** 4

**Summary:**

This paper proposes the Distil-E2D, which is a framework utilizing the prior knowledge from foundational depths models. This work utilizes DepthAnythingV2 to generate pseudo labels and complements sparse LiDAR depth maps. It introduces a Confidence-Guided Calibrated Depth Loss to align the synthetic and real depths and weight the confidence. This work additionally employs a Context Transformer and a Dual-Decoder Training scheme. Experiments also prove the good performance of the Distil-E2D.

**Questions:**

1. Some parts of Figure 1 are confusing: the arrow from DepthAnythingV2 to the “MDE Prediction dtm” looks insufficient, and according to the formula, the coefficient in $L_{\text{align}}$ between Calibrated Depth and LiDAR Depth should be $w_t^d$ rather than $W_c$.
2. In the abstract, the first occurrence of “MDE” should be accompanied by its full form, “monocular depth estimation.” Additionally, “MDE” and “EMDE” must be used consistently throughout the paper, many places use “MDE” (e.g., line 154, line 168) when it should actually read “EMDE”. References [28] and [37] are identical, and the authors should delete the duplicate one.
3. Why does Equation 7 use logarithmic depth instead of the raw depth? Is this based on a particular prior or intended to expand the effective depth range? Further explanation is needed.
4. Further clarification is required on:
   (a) the network architectures of the encoder and decoder;
   (b) for the MVSEC and DSEC datasets, considering poorly lit scenarios such as nighttime where the APS/RGB modality degrades severely, did the training set include only well-illuminated sequences?

**Ethical Concerns:**

["NO or VERY MINOR ethics concerns only"]

**Final Justification:**

The authors have addressed my concerns, and I think this solid work meets the acceptance requirements.  So I'm raising the overall score to 5.

**Limitations:**

yes

**Paper Formatting Concerns:**

This manuscript does not have any major formatting issues.

**Quality:**

4

**Strengths And Weaknesses:**

Strengths:
This paper focuses on monocular depth estimation with event cameras and provides thorough consideration and design in terms of pseudo labels, loss function formulation, and network architecture. The experimental results demonstrate the superiority of the proposed approach, and the ablation studies confirm the effectiveness of each module. The writing is clear, the content is easy to understand, and the work exhibits strong originality.

Weakness:
There are still some confusing aspects in the manuscript, which will be detailed below and require further clarification from the authors.

---

> ### Author Rebuttal · Authors · 2025-07-30
>
> **Q1: Inconsistencies in Figure 1 annotations and equations.**
>
> We thank the reviewer for pointing out the errors in the figure. The coefficient in $L_{align}$ between Calibrated Depth and LiDAR Depth should be $w^d_t$  instead of $W_c$. Furthermore, the arrow between DepthAnythingV2 and MDE Prediction should not exist. We will have the figure corrected in the camera-ready paper.
>
> **Q2: Inconsistent acronym usage and duplicate references.**
>
> We thank the reviewer for the feedback. We will align the naming and correct the references.
>
> **Q3: Justification for using logarithmic depth in Equation (7).**
>
> We use log depth instead of true depth because it offers numerical stability and better captures relative depth relationships, which is crucial when aligning synthetic and LiDAR depths. Depth values in driving scenes can span several orders of magnitude from very close objects (e.g., less than 2 meters) to distant background regions (tens of meters). Applying a log transform compresses this dynamic range into a bounded interval (approximately 0 to 1 in our normalized setup). This ensures that both near and far depths contribute comparably to the loss.
>
> **Q4: Lack of detailed encoder and decoder architecture descriptions.**
>
> Please refer to our response to Reviewer qS9Q Q2 for full architectural details, which will also be included in the final supplementary material.
>
> **Q5: Clarification on whether training data includes low-light sequences.**
>
> The training includes only well-illuminated sequences. For example,
> we use the `outdoor_day2` sequence in MVSEC for training, which is also the standard training set for other competing algorithms. We refrain from using nighttime sequences for training because of the degraded quality of monocular depth estimates in low light.

---

> ### Comment · Reviewer_jNtM · 2025-08-02
>
> Dear Authors:
>
> Thank you for your detailed rebuttal. I have carefully reviewed your responses to my previous comments and find that my concerns have been adequately addressed.
>
> I now regard the paper as a solid contribution to the field. I will accordingly increase my review score.
>
> Best regards

---

### Official Review · Reviewer_qS9Q · 2025-06-28

**Clarity:** 3
**Significance:** 3
**Originality:** 3
**Rating:** 5
**Confidence:** 4

**Summary:**

The missing of large volume and high-quality datasets for event-based tasks hinders the spreading of related applications such as Event Monocular Depth Estimation (EMDE).
For example, depth foundation models (VFM) such as Depth Anything V2 or MoGe leverage millions of high-quality labelled data to achieve astonishing results, while this kind of training is still unfeasible for the event domain, since high-quality-high-volume synthetic or real datasets are missing.

In particular with respect to EMDE, previous literature trained the models with real datasets such as MVSEC, where ground truth is collected using LiDAR, thus producing sparse and noisy labels.
This paper focuses on the high-quality part of the aforementioned problem, trying to overcome the problem by exploiting frame-based depth foundational models -- i.e., Depth Anything V2.

The core idea is to leverage aligned frame and event data -- e.g., using a DVS sensor that produces intrinsically aligned frame-event information -- to add an additional proxy-supervision signal from the estimations of the foundational model.
In other words, given an aligned frame, we can estimate an aligned proxy depth map using the foundation model and use it to complete the LiDAR information.

Additionally, the authors introduce a series of techniques to fuse LiDAR and proxy relative depth maps: 1) a MLP model to create a non linear depth mapping to convert the proxy labels to a metric scale; 2) a CNN confidence estimator model to weight the supervision of proxy labels; 3) a double head for metric and relative depth estimation.
The enhanced supervision is used to train a novel transformer-based architecture for both event-only monocular depth estimation and event-frame monocular depth estimation.

The authors validated the effectiveness of the proposal with two standard benchmarks -- i.e., MVSEC and DSEC -- with state-of-the-art performance.

**Questions:**

- Could you provide additional information regarding the network implementation as requested in the previous section?
- Could you confirm the inference times of MDDE+ and ERE models?
- Instead of using relative VFM, have the authors tried to use metric foundational models?

**Ethical Concerns:**

["NO or VERY MINOR ethics concerns only"]

**Final Justification:**

Given the Strengths and Weaknesses of the proposal, I give a score of "Accept". In my opinion, the event-based high-quality-high-quantity dataset issue is a huge problem for the event community that hinders its diffusion. The authors managed to handle the problem partially with an interesting solution. Furthermore, the authors replied to my concerns, thus confirming my positive score.

**Limitations:**

yes

**Quality:**

3

**Strengths And Weaknesses:**

Strengths

- **The paper is well written and easy to follow**: the authors present their methodology in a clear and structured way, with well-organized sections and illustrative figures that guide the reader through the key concepts.
- **Interesting idea to partially overcome to the event-data issue**: The paper addresses an important problem in the event-based literature -- i.e., missing high-quality-high-volume datasets -- with particular effort regarding the high-quality part. The proposed distillation framework is a novel paradigm to leverage the knowledge of frame-based VFM to increase the quality of event datasets.
- **SOTA performance**: The results show the effectiveness of the proposal against several competitors using two standard benchmarks -- i.e., MVSEC and DSEC.

Weaknesses

- **Fairness of the E+I experiments**: Given Fig. 2, the proposed Distil-E2D (E+I) Framework uses frozen DAv2 for estimation, thus leveraging (of course indirectly) millions of labeled images, while the other competitors do not. This raises concerns regarding the fairness of the E+I evaluation protocol.
- **Missing details about implementations**: I would like to see more details regarding the network implementation, since there are few details about it. For example, the size of the frozen Frame Encoder -- i.e., DAv2-S or DAv2-B or DAv2-L -- is missing. Also, there are no details about the encoders. I suggest that the authors include additional information, such as figures and implementation details in the supplementary.
- **Concerns about inference times**: I tried both MDDE+ and ERE models, but I found that MDDE+ is quite faster than ERE.
- **Usage of only one VFM for proxy-labelling**: The authors tried only one VFM for proxy supervision and Image Encoder. I suggest in future work to try other new VFM such as MoGe, DepthPro, and Lotus.
- **Missing study regarding the high-quantity dataset issue**: The authors missed the opportunity to study the effectiveness of the proposal regarding the high-quantity problem cited in the Summary section. In particular, the authors could have tried to increase the robustness of the network with a proxy pre-training using a large amount of aligned frame-event unlabelled data and proxy supervision from VFMs (leveraging the RDE head). I suggest this as future work for the authors.

Given the Strengths and Weaknesses of the proposal, I give an initial score of "Accept". In my opinion, the event-based high-quality-high-quantity dataset issue is a huge problem for the event community that hinders its diffusion.
The authors managed to handle the problem partially with an interesting solution.
Furthermore, I did not find any major weaknesses, thus confirming my positive score.

---

> ### Author Rebuttal · Authors · 2025-07-30
>
> **Q1: Fairness of using DAV2 in E+I experimental protocol.**
>
> We thank the reviewer for raising this point regarding the fairness of the E+I experiments. Using DAV2 as the  image encoder is a natural and principled design choice. It avoids the need to train a separate image encoder from scratch with limited data by using the rich priors of foundational depth models. Importantly, incorporating DAV2 does not trivially solve the depth estimation problem. APS frames often yield inaccurate depth predictions when relying solely on DAV2, particularly in challenging nighttime scenes. Additional challenges arise from depth misalignment between synthetic supervision and real-world data. In our E+I framework, events play a crucial role by offering high-temporal-resolution cues that complement DAV2 features. This leads to improved performance in scenarios where image-only models struggle. We thus view our approach as a meaningful extension of existing models instead of being an unfair advantage. This is because events remain essential to achieving accurate depth estimation.
>
>
> **Q2: Missing implementation details on encoder and DAV2 architecture.**
>
> We use DepthAnythingv2-L for the frame encoder in the E+I setting and for synthetic depth generation. The architecture of our event encoder, context encoder, and RDE/MDE decoder is as follows:
>
> **Event Encoder**
>
> | Module     | Operation                                                                 |
> |------------|---------------------------------------------------------------------------|
> | Input Head | Conv2d(5, 32, kernel size 5, stride 1, padding 2)                         |
> | Encoder 0  | Conv2d(32, 64, kernel size 5, stride 2) + ConvLSTM(128, 256, kernel size 3) |
> | Encoder 1  | Conv2d(64, 128, kernel size 5, stride 2) + ConvLSTM(256, 512, kernel size 3) |
> | Encoder 2  | Conv2d(128, 256, kernel size 5, stride 2) + ConvLSTM(512, 1024, kernel size 3) |
> | Encoder 3  | Conv2d(256, 512, kernel size 5, stride 2) + ConvLSTM(1024, 2048, kernel size 3) |
>
> **Context Encoder**
>
> | Module     | Operation                                         |
> |------------|---------------------------------------------------|
> | Input Head | Conv2d(6, 32, kernel size 5, stride 1, padding 2) |
> | Encoder 0  | Conv2d(32, 64, kernel size 5, stride 2)           |
> | Encoder 1  | Conv2d(64, 128, kernel size 5, stride 2)          |
> | Encoder 2  | Conv2d(128, 256, kernel size 5, stride 2)         |
> | Encoder 3  | Conv2d(256, 512, kernel size 5, stride 2)         |
>
> **Decoder (RDE and MDE Decoders)**
>
> | Module      | Operation                                           |
> |-------------|-----------------------------------------------------|
> | Decoder 0   | Upsample x2 + Conv2d(512, 256, kernel size 5, stride 1) |
> | Decoder 1   | Upsample x2 + Conv2d(256, 128, kernel size 5, stride 1) |
> | Decoder 2   | Upsample x2 + Conv2d(128, 64, kernel size 5, stride 1)  |
> | Decoder 3   | Upsample x2 + Conv2d(64, 32, kernel size 5, stride 1)   |
> | Output Head | Conv2d(32, 1, kernel size 1)
>
> Skip connections are applied between the event encoder and RDE/MDE decoders. We will include these architectural details in the supplementary material.
>
> **Q3: Inference time comparison between ERE and MDDE+.**
>
> We thank the reviewer for this observation. Upon revisiting our runtime measurements, we verified that
> runtime of ERE is as reported (35ms). However, we realized that our reported MDDE+ runtime (24.2ms) had mistakenly included pre-processing overhead. In our revised measurement, MDDE+ achieves approximately 8ms on our setup.
> This confirms that it is indeed much faster than ERE. We will update the runtime results accordingly.
>
> **Q4: Limited evaluation of proxy supervision using only one VFM.**
>
> We thank the reviewer for this insightful suggestion. Although our study focused on a single VFM to maintain a controlled analysis, exploring other VFMs such as MoGe, DepthPro, and Lotus, or even a mixture-of-experts design could potentially further improve depth distillation. We acknowledge this as a valuable direction for future work.
>
> **Q5: Scalability with large unlabelled datasets.**
>
> We agree that proxy pretraining on large-scale unlabelled event-image data is a highly promising direction. Our framework is inherently compatible with such scaling and has a strong potential to yield generalizable EMDE models when applied to large-scale data. Although the construction and processing of such datasets requires resources beyond the scope of this work, we view this as a natural and impactful extension of our approach and an important direction for future research.
>
> **Q6: Use of relative versus metric foundational models for depth supervision.**
>
> We ran some initial experiments with Depth Pro as a source of metric supervision. However, we found that the predicted depths often exhibited inconsistent global scaling and do not align well with the LiDAR ground truth across different scenes. In contrast, relative depth models such as Depth Anything V2 produced more temporally stable output with sharper object boundaries and more coherent segmentation of surfaces. These qualitative differences resulted in a better spatial structure, which we then calibrated to metric space using our NDC module. Our two-stage design, which first captures rich relative geometry before calibrating to metric space, offers a more reliable and modular alternative to directly relying on metric predictions from VFMs.

---

> > ### Comment · Reviewer_qS9Q · 2025-08-01
> >
> > Dear Authors,
> >
> > Thanks for the detailed response.
> > I acknowledge that I have read other reviews.
> >
> > I'm satisfied with the answers. In particular:
> >
> > - **Q2**: Please add the information on the architecture in the supplementary material: it is important for reproducibility.
> > - **Q2**: Just a little comment: the E+I model uses DepthAnythingv2-L, thus we will see a more significant drop in performance -- especially in DSEC, where the number of pixels is ~4x w.r.t. MVSEC. Have the authors tried DepthAnythingv2-S? Given Fig. 1 of DepthAnythingv2's paper, DepthAnythingv2-S is competitive w.r.t. Large, and at the same time ~4x faster.
> > - **Q3**: Thanks for the verification.
> > - **Q6**: Thanks for the insightful explanation.
> >
> > I will update my score at the end of the discussion.
> >
> > Best regards,
> >
> > qS9Q

---

> > > ### Author Response · Authors · 2025-08-04
> > >
> > > We are grateful to the reviewer for acknowledging the novelty and practical relevance of our approach.
> > >
> > > We also thank the reviewer for the insightful suggestion to consider DepthAnythingv2-S as the frame encoder in the E+I setting. To assess the tradeoff between performance and efficiency, we evaluated both DepthAnythingv2-L and DepthAnythingv2-S (abbreviated as DAV2-L and DAV2-S) within our E+I framework. The table below reports the **mean depth error at 30 meters** and **inference runtime per frame** on both MVSEC and DSEC:
> > >
> > > | Model Variant | MVSEC Error ↓ | DSEC Error ↓ | MVSEC Runtime (ms) ↓ | DSEC Runtime (ms) ↓ |
> > > |---------------|----------------|----------------|-------------------------|-------------------------|
> > > | DAV2-L        | **2.48**        | **1.61**        | 47.5                    | 167.1                   |
> > > | DAV2-S        | 2.57           | 1.75           | **25.2**                | **96.5**                |
> > >
> > > Our results confirm that DAV2-S offers a good efficiency advantage, reducing runtime by nearly half across both datasets. While there is a slight degradation in accuracy relative to DAV2-L, the performance remains relatively robust and the tradeoff is acceptable in settings where computational efficiency is a key consideration.

---

> > > > ### Comment · Reviewer_qS9Q · 2025-08-04
> > > >
> > > > Thanks for the nice additional experiment. I suggest that the authors also include this in the supplementary material.
> > > >
> > > > Best,
> > > >
> > > > qS9Q

---

### Official Review · Reviewer_Uz19 · 2025-07-02

**Clarity:** 3
**Significance:** 3
**Originality:** 2
**Rating:** 5
**Confidence:** 4

**Summary:**

This paper presents a model trained to predict dense depth maps from event-based cameras. It works by distillating priors about geometry of dense depth maps by using zero-shot depth prediction from a foundation model (DepthAnything-V2) on co-registered grayscale or RGB images that are "synchronized" with the event signals, gathered within a 3d block of event voxels over a fixed time interval.
To transform the relative depth maps predicted by DepthAnything-V2 to metric (absolute) depth maps, a calibration model is trained, supervised by LIDAR depth projected images, to learn a non linear calibration function. The calibrated synthesized dense depth maps are then used to supervise the event-base depth prediction model.
The proposed architectures uses a context branch fed by the past dense depth map prediction, that is combined to the input event data through a cross attention module performed by a transformer at the end of the decoder, before going through the decoder that outputs the dense depth maps.

**Questions:**

Could you observe or try to analyze the possible distortions induced by the context branch in terms of excessive dependence to the past (in particular in case of accelerations)?
Please introduce and make clear the APS acronym and the distinction with RBG images.
It is not clear BTW why the APS image appears as binary (saturated?) in Figure 1.
There is a notation problem in equation (5), since i is the sum index, it should be "The voxel value at (\mathbf{u},t)..." and "V(\mathbf{u},t) =" instead of (\mathbf{u}_i,t)

**Ethical Concerns:**

["NO or VERY MINOR ethics concerns only"]

**Final Justification:**

After reading the reviews and rebuttal, I am totally satisfied with the answers of the authors, and in particular to the slight concerns I had initially, I therefore maintain my "Accept" score.
Best regards,

**Limitations:**

One word of discussion about generalization and the possible strong dependence to the context could be added.

**Paper Formatting Concerns:**

No formatting concern has been detected.

**Quality:**

3

**Strengths And Weaknesses:**

The paper is clear and nicely written. The problem of estimating depth maps in nighttime or tunnel conditions is important so the subject is relevant. The proposed architecture and distillation setup make sense. The experimental part is well done, the results are good and the ablation study shows the benefits of the different components of the approach.
One possible weakness of the approach is the strong dependence to the context branch. One might wonder if this method could generalize to different contexts (e.g. handheld or aerial, with fast viewpoint changes), and how the past depth maps possibly distort (blur?) the current prediction.
Aside from this minor point I don't see major issue in this paper, except a few clarifications to add (see questions).

---

> ### Author Rebuttal · Authors · 2025-07-30
>
> **Q1: Potential distortions induced by the context branch under accelerated motion.**
>
> We analyze whether the context branch introduces distortions under acceleration on the MVSEC test set using event-only input. The test frames are segmented into low, medium, and high motion bins based on the average ground truth optical flow magnitude with thresholds defined by the 33rd and 66th percentiles. For each bin, we report the mean depth error at 30m.
>
> | Motion Bin | Baseline (no context branch) | Ours (w/ context branch) | $\Delta$ Delta |
> |-----------|-----------|-----------|-----------|
> | Low | 2.66 | $\textbf{2.34}$ | 0.32 |
> | Medium | 2.71 | $\textbf{2.52}$ | 0.19 |
> | High | 2.80 | $\textbf{2.69}$ | 0.11 |
>
> The results show that with the context branch, the accuracy decreases with increasing motion due to the growing misalignment between frames, which reduces the effectiveness of temporal fusion. Nonetheless, the context branch continues to provide valuable temporal cues and structural continuity, yielding improved accuracy over the baseline across low to high motion magnitudes.
>
>
> **Q2: Clarification on APS, its distinction from RGB, and the appearance of saturation in Figure 1.**
>
> APS (Active Pixel Sensor) frames are captured from the same imaging plane as events and are inherently co-aligned. In contrast, RGB frames are acquired from an external camera and require extrinsic calibration. As seen in Figure 1, APS frames may appear saturated under challenging lighting due to limited dynamic range. Although APS and events share the same imaging plane, they operate on distinct circuitry and sensing principles. Specifically, event data more robust to extreme illumination since APS measures absolute intensity while events encode log-intensity changes.
>
>
> **Q3: Notation error in Equation (5).**
>
> Thank you for highlighting this. We will correct the notation in the camera-ready version.
>
>
> **Q4: Discussion about model generalization.**
>
> Similar to existing EMDE models, our method currently does not generalize to open-world scenes and remains dataset-specific. This work focuses primarily on improving the quality of supervision through dense depth distillation. However, we acknowledge that the lack of large-scale event datasets remains a key bottleneck. We hope that our work motivates future efforts to acquire such datasets. This would enable our work and other future work to be extended to generalizable event-based monocular depth estimation.

---

> > ### Comment · Reviewer_Uz19 · 2025-08-01
> >
> > Dear authors,
> >
> > Thank you for your detailed answers to all reviewer's comments.
> >
> > I am totally satisfied with the response you made to my own (slight) concerns by:
> > 1) Providing experimental results with different accelerations to assess the dependence to the context branch
> > 2) Clarifying the distinction between APS sensors and RGB cameras and its relation to this work
> > 3) Mentioning that the dependence to the context is not expected to be less than other MDE methods.
> >
> > Best regards,

---

### Note · Authors · 2025-08-12

We sincerely thank all reviewers for their time and constructive feedback on our work. We appreciate that the reviewers recognized the novelty and potential impact of our contributions. Your comments have helped us address potential limitations and improve the presentation of our manuscript. We have carefully considered all suggestions and will incorporate the necessary revisions to further strengthen the technical rigor and clarity of Distil-E2D.

---

### Decision · Program_Chairs · 2025-09-17

**Decision:**

Accept (poster)

**Comment:**

The final ratings are 3 accepts and 1 borderline accept. The AC have read the reviews and rebuttal, and discussed the submission with the reviewers. The reviewers agreed that the paper presents an interesting idea to overcome data limitations. Nonetheless, they raised a number of points during the review phase including fairness of experiments, lack of implementation details, computational complexity, and clarity of presentation. The authors were able to clarify these points during the rebuttal and discussion phases. The AC recommends the authors to incorporate the feedback and suggestions provided by the reviewers, and the materials presented in the rebuttal, which would improve the next revision of the manuscript.